# Assessment of Postpartum Stress Using the Maternal Postpartum Stress Scale (MPSS) in Spanish Women

**DOI:** 10.3390/healthcare12101032

**Published:** 2024-05-16

**Authors:** Sergio Martínez Vázquez, Adrián Ruíz Perete, Alejandro de la Torre-Luque, Sandra Nakić Radoš, Maja Brekalo, Carmen Amezcua-Prieto, Rafael A. Caparros-Gonzalez

**Affiliations:** 1Campus Las Lagunillas s/n, University of Jaén, 23071 Jaén, Spain; 2Faculty of Medicine, University Complutense of Madrid, 28040 Madrid, Spain; adriru06@ucm.es (A.R.P.); af.delatorre@ucm.es (A.d.l.T.-L.); 3Centro de Investigación Biomédica en Red de Salud Mental (CIBERSAM), Instituto de Salud Carlos III (ISCIII), 28029 Madrid, Spain; 4University Department of Psychology, Catholic University of Croatia, 10 000 Zagreb, Croatia; snrados@unicath.hr (S.N.R.); maja.brekalo@unicath.hr (M.B.); 5Department of Preventive Medicine and Public Health, Faculty of Medicine, University of Granada, 18016 Granada, Spain; carmezcua@ugr.es; 6Instituto de Investigación Biosanitaria ibs GRANADA, 18071 Granada, Spain; rcg477@ugr.es; 7Consortium for Biomedical Research in Epidemiology and Public Health (CIBERESP), 28029 Madrid, Spain; 8Department of Nursing, University of Granada, 18016 Granada, Spain

**Keywords:** maternal postpartum stress scale (MPSS), Spanish women, surveys and questionnaires

## Abstract

Although scales that evaluate postpartum stress exist, they lack specificity in maternal postpartum stress. The MPSS was created because there was a need to assess maternal stress during the postpartum stage. The introduction of the MPSS has enriched the evaluation tools for postpartum stress and has helped understand maternal stress at various postpartum time points and identify women at high risk for postpartum stress during this period. The aim was to translate the MPSS into Spanish and study its psychometric properties. Postpartum women (N = 167) with a mean age of 34.26 (SD = 4.71) were involved in this study. In addition to the MPSS, a battery of instruments was administered: a demographic sheet, the Birth Satisfaction Scale-Revised (BSS-R) and the Edinburgh Postnatal Depression Scale (EPDS). The MPSS data were analyzed, checking item communality first. As a result, three items showed unsatisfactory communality values (h^2^ < 0.40). Confirmatory Factor Analysis was conducted, comparing factor models using the full pool of MPSS items or the version without items with unacceptable communality. As a result, the original three-factor structure was endorsed on the Spanish MPSS, with better fit indices when removing items with low communality (RMSEA = 0.067, CFI = 0.99, TLI = 0.99). The reliability of this version was satisfactory (ω = 0.93). Finally, group comparisons for some perinatal variables were performed, showing no significant differences between groups of interest (*p* = 0.05 and above). To conclude, the MPSS will contribute to the existing literature, having a wider capacity to assess perinatal mental health difficulties in Spanish-speaking populations.

## 1. Introduction

Postpartum is a critical period for both the mother and child, and it can have negative effects on maternal health [1,2,3]. Postpartum stress is a common psychological response to the responsibilities of raising a child, and it often leads to recurrent and sometimes unrealistic concerns and fears about several aspects of motherhood and personal health [1].

It is worth mentioning the influence of genetic, biological, and hormonal factors on the risk of mental disorders in the perinatal period. However, it is equally important to incorporate psychosocial factors, including the impact of stressors throughout a person’s life, when recognizing these disorders [4]. 

Furthermore, the undeniable connection between maternal mental health and the well-being of both mothers and their infants should be considered due to its influence on pathophysiological mechanisms for mental disorder development [5,6]. The evidence supporting the effects of maternal stress during pregnancy on adverse neurodevelopmental outcomes for the child is extensive [7,8,9]. This phenomenon could be a result of “fetal programming”, which has been widely studied and documented [10]. Furthermore, maternal stress during pregnancy has been linked to behavioral and emotional problems, as well as decreased gray matter density in childhood [7]. Maternal stress can eventually lead to disabling mental conditions, such as postpartum depression or posttraumatic stress disorder [2,11,12] and poor mother–infant bonding [2,11,12,13]. The effect on maternal health may be a significant contributor to maternal mortality via suicide behavior [4,14].

Advocating for mental well-being and protecting the mental health of this population is not just a responsibility but an obligation [4]. By covering maternal mental health needs, the overall health and vitality of families as a whole may be ensured [15,16,17,18]. 

Postpartum stress often goes unrecognized because it is associated with specific mental conditions, making it difficult to identify, and interventions to tackle postpartum stress may not be tailored and efficient [19,20]. While structured interviews are commonly used to detect postpartum stress, they are time-consuming and require trained professionals [1,2]. On the contrary, this appears to be intertwined with women’s need to tell their stories in detail, to be heard and validated. Experiencing trauma during the existential experience of birth requires a distinctly different approach than other traumas [21]. Self-report measures may be more efficient in field studies and accessible for screening and early detection of postpartum stress [3]. 

In 2023, the Maternal Postpartum Stress Scale (MPSS) was developed to assess postpartum stress during the postpartum period [2]. This scale has enriched the assessment tools for postpartum stress and can help identify women at high risk for postpartum stress, a property that no other scale possesses. However, the MPSS has not been validated in other ethnic groups different from the Chinese population [22]. Therefore, this study aims to translate and culturally adapt the MPSS into a Spanish-speaking context and analyze its psychometric properties in a group of Spanish-speaking postpartum women.

## 2. Materials and Methods

A cross-sectional observational study in which sociodemographic and some labor-related variables were collected and then descriptively analyzed using mean and SD or percentage of cases when applicable. The inclusion criteria consisted of women between 18 and 45 years old who have given birth recently (according to postnatal checks) and attended the postpartum visit in the primary care centers of the Granada-Metropolitan Health District. The women were approached by midwives and other healthcare professionals during the visit. Due to the recruitment process (captured when attending postpartum checks) and briefing by their midwives prior to completing the questionnaire, the sample can be considered representative of the postnatal women population. 

The MPSS (Maternal Postpartum Stress Scale) [2] is a reliable tool for assessing self-reported postpartum stress over the previous month with good validity evidence. It is made up of 22 items divided into three subscales: personal needs and fatigue, infant nurturing and body changes, and sexuality. The personal needs and fatigue dimension comprises nine items related to adjustment to wake-ups, the baby’s sleep patterns, fatigue, household chores, the lack of help and time for socializing and oneself, the inability to complain, and feelings of loneliness. The infant nurturing dimension includes seven items related to feeding, the baby’s development and health, recognizing the baby’s needs, and soothing a crying baby. The body changes and sexuality dimension consists of six items related to uncertainty about resuming intercourse, frequency and enjoyment of sexual intercourse, feelings of unattractiveness and difficulty returning to pre-pregnancy weight and appearance. The participant rates each item on a 5-point Likert scale, ranging from 0 (not experiencing at all) to 4 (experiencing completely). The higher the score on the MPSS, the higher the levels of postpartum stress are. 

The Edinburgh Postnatal Depression Scale ((EPDS); [23] measures depressive symptoms covering cognitive and emotional symptoms over the last week in postpartum mothers. Ten items are rated on a 4-point scale (0 to 3). The total score ranges from 0 to 30 where a higher score signifies higher depressive symptoms. The scale was originally adapted and validated in Spanish postpartum women with a sensitivity of 79% and specificity of 95.5% being its negative predictive value of 97.7% [24]. The McDonald’s ω in the current study was 0.89. 

The Birth Satisfaction Scale-Revised (BSS-R) is a self-measuring instrument designed by Hollins Martin and Martin [25] and comprises 30 items. Answers are scored on a 5-point Likert scale, with 1 representing the lowest satisfaction and 5 representing the highest satisfaction. A higher score represents higher satisfaction with the birth experience. This scale was applied and adapted to a Spanish context with internal reliability found to be acceptable (α, >0.70). Evidence for good divergent and convergent validity was also found across total and sub-scale scores; for divergent validity, total and sub-scale scores and participant age showed good correlations (S-BSS-R total score r = −0.06, *p* = 0.36, SE r = −0.06, *p* = 0.43, WA r = −0.04, *p* = 0.54, QE r = −0.05, *p* = 0.46), and for convergent validity, positive correlations were observed between S-BSS-R total, SE and QC sub-scores and the PSS (r = −0.20, *p* = 0.006, r = −0.20, *p* = 0.006 and r = 0.14, *p* = 0.05, respectively) [26]. 

The demographic sheet included demographic questions on women’s age, education, relationship status and income and obstetrical information about the mode of birth, medical complications during pregnancy and childbirth for the mother or the child, support from the partner during labor (low/enough/a lot), previous pregnancy loss and their traumatic perception of birth (0–10).

It was planned to have five participants per item [27], making it necessary to have at least 110 participants. Although the sample size was recommended to be at least 110 women, we decided to recruit as many women as possible to improve the statistical analysis. The MPSS items were analyzed to check the distribution features, and normality and communality were calculated. The reliability of the scale and the subscales were estimated by using omega and alpha, respectively. Then, Confirmatory Factor Analysis (CFA) was performed using Diagonally Weighted Least Squares (DWLS) as an estimation method, with the Satorra–Bentler correction for the χ^2^ applied. The fit of four confirmatory different models was compared: two unidimensional and two with three dimensions, two with all the items, and two excluding items with communality < 0.40. In the CFA, there were considered the fit indices CFI, TLI (for both indexes > 0.95 to consider an adequate fit), RMSEA and SRMR (both indexes < 0.08), as well as the Pearson χ^2^. Finally, group comparisons for the labor-related variables were performed: *t*-test, analysis of variance (ANOVA), and Spearman correlation, depending on each variable. The data were analyzed using R software version 4.3.2., including lavaan and psych packages.

For the collection of information, approval has been obtained from the Research Ethics Committee of the Biomedical Research of the province of Granada ((CEIM/CEI GRANADA) code: 0880-N-21). All participants had to read the instructions and freely accept an informed consent before beginning to complete the questionnaire. The data were handled in a disaggregated manner, and the anonymity of the participants was maintained at all times. 

## 3. Results

The sample consisted of 167 women between 21 to 49 years old (*M* = 34.26, *SD* = 4.71), most of them married or in civil partnership (65.27%) or living with a partner (32.93%) and with higher education (78.44%), mostly with average (70.06%) or above average (17.96%) income. Table 1 displays the distribution of participants for each option for some other relevant population variables, which will be used for posterior analysis.

Table 2 displays the descriptives of the MPSS items. Communality was lower than 0.4 for the three following items: MPSS3, MPSS5 and MPSS7. This shows the percentage of the variance of each item explained by the factors of the scale, which is desired to be as good as possible since it means the items are appropriately explained by the factors. The opposite would lead to a loss of internal consistency in the scale. For this reason, the performance of those items is going to be analyzed.

To continue this analysis, four models were estimated and compared by Confirmatory Factor Analysis (CFA). The first was a model with one factor and all items. The second model was also a model with one factor but taking out the three items with communality < 0.4. The third one was a three-factor model with all the items. The final model was also a three-factor model but without the three items. Fit indices for the models are presented in Table 3.

As can be seen in Table 3, both one-factor and three-factor models performed very similarly, with the three-factor models fitting better to data because they showed adequate values in the fit indices, which was to be expected. In the three-factor models, the one without the three items showed values in both CFI and TLI slightly better than the model with all the items, as well as a very slightly reduced SRMR, only facing an equally minuscule increase in RMSEA value, still maintaining acceptable values. 

To provide further evidence to keep or remove the items with low communality within the structural model, we checked the change in reliability if they were removed. For the whole set of items, the general reliability, measured with the McDonald’s ω total, was 0.94, while the reliability of the subscales, measured with α, was 0.84 for the infant nurturing subscale, 0.89 for the personal needs and fatigue subscale and 0.85 for the body changes and sexuality subscale, which prove to be adequate results.

If the mentioned items were taken out, the general reliability would remain unchanged, as well as the other two subscales, since their items would remain unchanged. It would be in the first subscale, infant nurturing, where the change would happen, increasing its α to 0.86. As this shows, those three items were items that were most likely not working well and even hindered the reliability of the subscale they were part of.

Shifting to a different aspect of our analysis, the subsequent section will delve into the discriminant validity via known-group differences and comparison of groups concerning labor-related variables. The variables used for the comparisons are the ones included in Table 1. The results for the *t*-test are presented in Table 4 and for the ANOVA in Table 5. None of the variables showed significant differences between the groups. The correlations between the traumatic perception of birth and the total and subscales scores were also found to be non-statistically significant, ranging from −0.15 to −0.02. In other words, levels of maternal postpartum stress were equal irrespective of the mode of birth, level of support during labor, medical complications during pregnancy and childbirth, having had a pregnancy loss or the levels of traumatic perception of birth. The correlation between age and the MPSS scores was also calculated, showing no significant correlation between age and total, IN and PNF subscale score, but a significant and negative correlation (ρ = −0.17; *p* = 0.02) with the BCS subscale. 

## 4. Discussion

Maternal postpartum stress is an important factor influencing the physical and mental well-being of women after childbirth. Existing scales for assessing postpartum stress are not tailored to the specific needs of new mothers and are generally universal in nature, thus considering maternal stress as a dimension of a complex tool. Furthermore, there is a lack of tools available that comprehensively measure postpartum stress from various perspectives. To address this gap, the Maternal Postpartum Stress Scale (MPSS) was developed in Croatia [2], specifically for women within the first year of childbirth. In this study, the researchers aimed to translate, adapt and validate the MPSS to use it in Spanish-speaking women. Some evidence of validity was provided by checking the descriptive statistics of the items and communality, performing a CFA and reliability analysis and ending with group comparisons on relevant variables.

The researchers assessed the psychometric properties of the Spanish version of the MPSS through a cross-sectional study with postpartum women. The general reliability proved to be great. This goes along the line of the results obtained in the Chinese version [22]. In terms of the reliability of the sub-scales coming from the original version, the alpha statistic for the three dimensions was lower than in Wang, Gao et al.’s [22] validation. However, this value is still considered to show good reliability for all of the dimensions [28].

Conversely to other versions of the MPSS, the items MPSS3, MPSS5, and MPSS7 showed a low communality. It was seen that if they were taken out, the general reliability would remain unchanged, but the first subscale’s reliability would increase to 0.86. The other two subscales would not change since their items would remain the same. 

With these results, considering that the communality of items 3, 5, and 7 is below the recommendation of 0.4, and the reliability of their subscale increases if they are left out, while the CFA model maintains adequate levels but the model without those items had similarly satisfactory fit indices, our recommendation would be to not include these items on the questionnaire. However, since the sample is relatively small, and considering how small the shown benefits of excluding these items are (not to mention that some others were on the verge of the 0.4 communality threshold), it would be reasonable to pursue these flaws in future investigations, as to show greater light in the topic and have greater confidence in the results.

As this shows, those three items were most likely not working well and even hindering the reliability of the subscale they were part of. This may be taken into consideration when using the scale looking for an association between variables associated with postpartum stress [3]. Thus, as this is caused by low communality on these items, recommendations made by other authors were followed [29].

Furthermore, the group comparisons conducted via ANOVA and *t*-test analysis did not show any significant differences between the MPSS score and the variables mother complications, infant complications, pregnancy loss, type of delivery or partner support. Homogeneity in a sample is critical in validation studies because it ensures that the results are not skewed by differences in the characteristics of the participants. ANOVA and *t*-test analysis can help researchers determine if there are significant differences between groups, which can help validate the findings and ensure the reliability of the study results [30].

The group comparisons were performed to study differences between these variables and the three MPSS dimensions and total scores. The correlations were found to be non-statistically significant, ranging from −0.15 to −0.02. Conversely to our findings, Wang et al. [3] found associations between age, education levels (including paternal) and high BMI with all the dimensions of the MPSS. This may be due to sample size differences (167 vs. 406) or the specificity of the population of the study (Spanish-speaking women). 

Moreover, these authors found that the associations vary during the postpartum phase depending on the month and the dimension, considering a higher BMI a determining factor 6 months after delivery affecting body changes and sexuality dimension [3]. Lewis et al. [31] found in a randomized trial that exercise can help to reduce maternal perceived stress at 6 months postpartum. Considering that regular exercise can help to reduce BMI, a significant reduction in perceived stress can be granted by advocating women’s exercise interventions. 

The American College of Obstetricians and Gynecologists released new recommendations in 2023 with the aim of improving the overall well-being of pregnant and postpartum women and families. A key message conveyed in the accompanying discussion papers emphasized the necessity of going beyond the mere identification of perinatal illness. Instead, it urged practitioners to proactively screen women who are at risk for postpartum psychiatric illness [32]. Ultimately, the focus was on the comprehensive treatment and holistic assistance required for these vulnerable populations [33]. The MPSS will help with the screening of Spanish-speaking women due to its properties. Considering the Spanish-speaking countries as well as the worldwide population that speaks the language, potentially more than 20 different countries and several continents may benefit from it [34]. The MPSS should be culturally adapted when applying this measure to an American Spanish-speaking country that is different from Spain. Cross-cultural adaptations of psychological measures should follow the International Test Commission guidelines [35]. 

The validation of the MPSS in Spanish-speaking women can help healthcare providers identify women who are at risk for postpartum stress and provide them with appropriate support and interventions. By screening women for postpartum stress using the MPSS, healthcare providers can ensure that these women receive the care they need to prevent further negative outcomes, such as postpartum depression or anxiety. In addition, it provides researchers with a reliable and valid tool to measure postpartum stress in Spanish-speaking populations. This can help researchers conduct more rigorous and accurate studies on postpartum stress, allowing for better comparisons between different populations and improving our overall understanding of this important issue. Furthermore, in case no specific language translation is available for the instrument, further investigations should focus on the usage and measurement of maternal depression with instruments such as the Patient Health Questionnaire (PHQ-9) [36].

However, the study has limitations, including a limited sample from a specific region in Spain. Future studies should include participants from different regions and explore other factors that may influence maternal stress. Additionally, longitudinal studies are needed to examine postpartum stress at various time points. Despite these limitations, the study provides valuable insights for future research and clinical practice related to maternal postpartum stress.

## 5. Conclusions

The MPSS is a comprehensive and culturally relevant tool specifically designed to assess postpartum stress in mothers during the first postpartum year. This study adds to the applicability in different languages, including Spanish-speaking women. Unlike other existing tools that may not fully capture the unique experiences and stressors faced by this population, the MPSS takes into account cultural differences in the expression and experience of stress. This customizability makes it a superior choice for healthcare professionals working with Spanish-speaking women, as it allows for a more accurate assessment of postpartum stress and the development of targeted interventions that are culturally sensitive and effective. By utilizing the MPSS, healthcare professionals can better support Spanish-speaking women during the crucial postpartum period, ultimately leading to improved mental health outcomes for both mother and baby.

## Figures and Tables

**Table 1 healthcare-12-01032-t001:** Sample characteristics (N = 167).

Item	Values
How was the baby born	
Vaginal birth	99 (59.28%)
Assisted vaginal birth	29 (17.37%)
Cesarean section	39 (23.35%)
Medical complications in mother during this pregnancy/childbirth	
Yes	61 (36.53%)
Medical complications in children during pregnancy/childbirth	
Yes	32 (19.16%)
Support from partner during labor	
Low support	23 (13.77%)
A lot	47 (28.14%)
A great deal	97 (58.08%)
Ever experienced pregnancy loss	
Yes	28.14%
Traumatic perception of birth	
Range	0–10
Mean	2.75
SD	2.91

Note. Proportion of cases is provided for all the variables except the last one.

**Table 2 healthcare-12-01032-t002:** Description of the MPSS items (N = 167).

Item	M (SD)	Skew	Kurtosis	h^2^
MPSS1	1.6 (1.3)	0.3	−1.1	0.8
MPSS2	1.5 (1.1)	0.5	−0.5	0.78
MPSS3	1.0 (1.3)	0.9	−0.4	0.35
MPSS4	1.4 (1.1)	0.7	−0.2	0.69
MPSS5	1.4 (1.1)	0.7	−0.3	0.37
MPSS6	1.4 (1.1)	0.5	−0.7	0.40
MPSS7	1.4 (1.1)	0.5	−0.6	0.33
MPSS8	1.5 (1.1)	0.5	−0.4	0.40
MPSS9	1.6 (1.0)	0.4	−0.2	0.53
MPSS10	1.8 (1.1)	0.3	−0.7	0.42
MPSS11	2.0 (1.2)	0	−1.0	0.42
MPSS12	1.6 (1.2)	0.4	−0.7	0.46
MPSS13	1.2 (1.3)	0.8	−0.6	0.52
MPSS14	1.3 (1.2)	0.7	−0.5	0.79
MPSS15	1.1 (1.3)	1.0	0.0	0.66
MPSS16	1.1 (1.3)	0.9	−0.3	0.57
MPSS17	1.3 (1.3)	0.7	−0.6	0.45
MPSS18	1.3 (1.2)	0.7	−0.3	0.69
MPSS19	2.1 (1.2)	0.2	−1.1	0.77
MPSS20	1.5 (1.2)	0.6	−0.5	0.57
MPSS21	1.2 (1.2)	0.9	0.0	0.72
MPSS22	1.4 (1.3)	0.6	−0.9	0.63

Note. h^2^ is the communality for each item.

**Table 3 healthcare-12-01032-t003:** Fit indices of the different CFA models.

Model	χ^2^	df	CFI	TLI	RMSEA (CI)	SRMR
1 factor—all items	1087.942	209	0.917	0.908	0.113 (0.106–0.120)	0.125
1 factor—minus 3 items	885.257	152	0.92	0.91	0.118 (0.111–0.126)	0.128
3 factors—all items	504.564	206	0.991	0.99	0.065 (0.058–0.072)	0.082
3 factors—minus 3 items	393.125	149	0.992	0.991	0.067 (0.059–0.075)	0.081

Note. All models were significant. df: degrees of freedom. CFI: Comparative Fit Index. TLI: Tucker–Lewis Index. RMSEA: Root Mean Square Error of Approximation. SRMR: Standardized Root Mean Squared Residual. χ^2^ corrected by Satorra–Bentler correction. CI: 90% confidence interval.

**Table 4 healthcare-12-01032-t004:** Results of the *t*-test.

	Absence of Complications/Loss	Presence of Complications/Loss	*t* (df)	*p*	Cohen’s *d*
	M (SD)	M (SD)			
Mother complications					
Total	30.89 (15.78)	33.02 (15.6)	−0.84 (165)	0.4	0.14
IN	9.27 (6.04)	10.34 (5.66)	−1.13 (165)	0.26	0.18
PNF	14.15 (7.56)	15.18 (8.01)	−0.83 (165)	0.41	0.13
BCS	7.46 (5.82)	7.49 (5.67)	−0.03 (165)	0.97	0.01
Infant complications					
Total	32.01 (16.26)	30.19 (13.2)	0.59 (165)	0.56	0.12
IN	9.63 (5.95)	9.81 (5.84)	−0.16 (165)	0.88	0.03
PNF	14.66 (7.91)	13.97 (6.91)	0.45 (165)	0.65	0.09
BCS	7.73 (5.87)	6.41 (5.17)	1.17 (165)	0.24	0.23
Pregnancy loss					
Total	32.35 (16.48)	29.91 (13.52)	0.98 ^†^ (101.77)	0.33	0.16
IN	9.37 (5.84)	10.43 (6.07)	−1.04 (165)	0.3	0.18
PNF	15.15 (8.08)	12.94 (6.51)	1.84 ^†^ (103.67)	0.07	0.29
BCS	7.83 (6.28)	6.55 (4.02)	1.56 ^†^ (130.12)	0.12	0.22

Note. None of the comparisons were statistically significant. ^†^: Welch two-sample *t*-test was used. IN: infant nurturing. PNF: personal needs and fatigue. BCS: body changes and sexuality.

**Table 5 healthcare-12-01032-t005:** Results of the ANOVA.

	1	2	3	F(2, 164)	*p*	η^2^
	M (SD)	M (SD)	M (SD)			
Type of delivery	*Vaginal birth*	*Assisted vaginal birth*	*Cesarean section*			
Total	33.38 (16.19)	30.34 (14.35)	28.28 (15.11)	1.61	0.2	0.02
IN	9.91 (6.11)	10.41 (5.42)	8.49 (5.71)	1.1	0.34	0.01
PNF	15.27 (7.58)	13.07 (7.17)	13.72 (8.37)	1.2	0.31	0.01
BCS	8.2 (6.19)	6.86 (5.08)	6.08 (4.76)	2.14	0.12	0.03
Partner support	*Low support*	*A lot of support*	*A great deal*			
Total	29.35 (17.15)	33.02 (15.57)	31.56 (15.51)	0.43	0.65	<0.01
IN	9.17 (6.76)	9.49 (5.14)	9.87 (6.09)	0.16	0.86	<0.01
PNF	13.87 (7.7)	15.77 (8.31)	14.08 (7.43)	0.85	0.43	0.01
BCS	6.3 (5.35)	7.77 (5.62)	7.61 (5.92)	0.56	0.57	<0.01

Note. IN: infant nurturing. PNF: personal needs and fatigue. BCS: body changes and sexuality. F: Snedecor’s F. η^2^: eta-squared.

## Data Availability

The original contributions presented in the study are included in the article, further inquiries can be directed to the corresponding author.

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
