# Peer review of "Assessment of Postpartum Stress Using the Maternal Postpartum Stress Scale (MPSS) in Spanish Women"

_healthcare, 2024, doi:10.3390/healthcare12101032_

Round 1
Reviewer 1 Report
Comments and Suggestions for Authors
1 please recheck the keywords by using Mesh Browser
2. Introduction part: I am not sure about cultural adaptation because it was conducted only in Spain. the duration of "Maternal leave" has been different across countries. So how many days in Spain? The difference in maternal leave according to the ILO should be discussed in the discussion part.
3. Kindly provide validities of measurements in Spanish for all measurements.
4. Please suggest further investigations of the usage and the measurement for maternal postpartum depression in case no specific language is translated., for example, Srisurapanont M, Oon-Arom A, Suradom C, Luewan S, Kawilapat S. Convergent Validity of the Edinburgh Postnatal Depression Scale and the Patient Health Questionnaire (PHQ-9) in Pregnant and Postpartum Women: Their Construct Correlations with Functional Disability. Healthcare (Basel). 2023 Feb 27;11(5):699.
5. Spanish-speaking countries span across 20 different countries and continents. Therefore, the measurements should be valid in a greater number of countries
6. Please update the references to be during 2014-2024. Some references are outdated.
Author Response
16 April 2024
Dear Editor of Healthcare,
Thank you very much for the opportunity to revise and improve the Manuscript ID: healthcare-2948227 entitled “Assessment of Postpartum Stress using the Maternal Pospartum Stress Scale (MPSS): Psychometric properties in Spain”.
First, we want to sincerely thank the reviewers for their constructive comments. Their comments have allowed us not only to significantly improve the manuscript but also to reflect on future research in the field of women´s health.
Following this letter, we detail point-by-point our responses to the reviewers' suggestions and the changes we have made in the revised version of our manuscript. We have highlighted the modifications made to the original text in the revised manuscript. We hope that the improvements made, along with the associated responses below, are sufficient for the definitive approval of the Editorial Team of Healthcare. If not, all authors remain at your disposal to resolve any issue.
We look forward to hearing from you at your earliest convenience.
Sincerely,
Sergio Martínez Vázquez
Reviewer: 1
Comments to the Author
1 please recheck the keywords by using Mesh Browser
Author´s Response: Thank you very much for your comments and suggestions that have helped to improve the manuscript. All the keywords have been checked and they appears now as: Maternal Postpartum Stress Scale (MPSS); Spanish Women (https://meshb.nlm.nih.gov/record/ui?ui=D014930), Surveys and Questionnaires (https://meshb.nlm.nih.gov/record/ui?ui=D011795). Please note that MPSS is not a keyword appearing in the Mesh Browser; However as it stands for the main aim of the manuscript we think is appropriate to establish it as a keyword.
- Introduction part: I am not sure about cultural adaptation because it was conducted only in Spain. the duration of "Maternal leave" has been different across countries. So how many days in Spain? The difference in maternal leave according to the ILO should be discussed in the discussion part.
Author´s Response: Thank you very much for your comments and suggestions. We corrected as can be read now as postpartum period, not linked exclusively to maternity leave. We had it well described in Methods section so we rewrite it both statements to be accurately the same as can be seen now.
- Kindly provide validities of measurements in Spanish for all measurements.
Author´s Response: Thank you very much for your comments and suggestions. We added Spanish validation and properties for the instruments used; The Edinburgh Postnatal Depression Scale (EPDS) (The scale was validated in Spanish postpartum womenwith a sensitivity of 79% and specificity of 95.5% being its negative predictive value of 97.7%. (Garcia Esteve et al., 2003) Lines 114-116); The Birth Satisfaction Scale (BSS); This scale was used and adapted to Spanish context with internal reliability found to be acceptable. Evidence for good divergent and convergent validity was also found across total and sub-scale scores (Romero-Gonzalez et al., 2019) Lines 121-128.
- Please suggest further investigations of the usage and the measurement for maternal postpartum depression in case no specific language is translated., for example, Srisurapanont M, Oon-Arom A, Suradom C, Luewan S, Kawilapat S. Convergent Validity of the Edinburgh Postnatal Depression Scale and the Patient Health Questionnaire (PHQ-9) in Pregnant and Postpartum Women: Their Construct Correlations with Functional Disability. Healthcare (Basel). 2023 Feb 27;11(5):699.
Author´s Response: Thank you very much for your comments and suggestions. We added the need for further investigations of the usage and the measurement for maternal postpartum depression in case no specific language is translated such as Patient Health Questionnaire (PHQ-9) by adding the cite provided.
- Spanish-speaking countries span across 20 different countries and continents. Therefore, the measurements should be valid in a greater number of countries
Author´s Response: Thank you very much for your comments and suggestions. We completely agree with you; thus we added a couple of lines to develop deeper this benefitial effect of the scale: Considering the Spanish-speaking countries as well as worldwide population that speaks the language, potentially more than 20 different countries and several continents may benefit from it (Statista Research Department, 2023). Lines 283-289.
- Please update the references to be during 2014-2024. Some references are outdated.
Author´s Response: Thank you very much for your comments and suggestions that have helped to improve the manuscript. We added a few references with publishing date between the period from 2014-2024. We think with this addition, the sum of references is well uptodated and adjusted to the topic.
Reviewer 2 Report
Comments and Suggestions for Authors
This paper presents the psychometric properties of a new test in Spain. The major points that need to addressed by the authors are the following:
The title is misleading as only preliminary can this paper be.
How was the sample size estimated? Please add this info.
Why was the accompanying tests administered? If they are relevant why not other tools been used instead of them. Please justify.
Authors need to present in a more detailed way the relevant research in other cultural contexts both quantitative using the same tool as well as qualitative research relevant to perspectives to this topic (e.g. doi: 10.4081/hpr.2020.9178).
Why did you choose this specific age range?
Is the sample representative? Please provide detailed information about the recruitment and testing process.
The statistical analyses are naive. Please read relevant papers presenting new methods in psychometrics (https://doi.org/10.1037/met0000317)
The discussion must be more critical and include more relevant references.
Finally, the conclusions need to be more extended and mention in a clear way why this tool is better than other existing tools for use in the Spanish population of women.
Comments on the Quality of English LanguageModerate English language editing.
Author Response
16 April 2024
Dear Editor of Healthcare,
Thank you very much for the opportunity to revise and improve the Manuscript ID: healthcare-2948227 entitled “Assessment of Postpartum Stress using the Maternal Pospartum Stress Scale (MPSS): Psychometric properties in Spain”.
First, we want to sincerely thank the reviewers for their constructive comments. Their comments have allowed us not only to significantly improve the manuscript but also to reflect on future research in the field of women´s health.
Following this letter, we detail point-by-point our responses to the reviewers' suggestions and the changes we have made in the revised version of our manuscript. We have highlighted the modifications made to the original text in the revised manuscript. We hope that the improvements made, along with the associated responses below, are sufficient for the definitive approval of the Editorial Team of Healthcare. If not, all authors remain at your disposal to resolve any issue.
We look forward to hearing from you at your earliest convenience.
Sincerely,
Sergio Martínez Vázquez
Reviewer: 2
Comments to the Author
This paper presents the psychometric properties of a new test in Spain. The major points that need to addressed by the authors are the following:
Author´s Response: Thank you very much for your comments and suggestions that have helped to improve the manuscript.
The title is misleading as only preliminary can this paper be.
Author´s Response: Thank you very much for your comments and suggestions. We have changed it and now reads as “Assessment of Postpartum Stress using the Maternal Pospar-tum Stress Scale (MPSS) in Spanish Women”.
How was the sample size estimated? Please add this info.
Author´s Response: Thank you very much for your comments and suggestions. It was already in the manuscript, however we added a couple of lines to clarify this aspect.
Why was the accompanying tests administered? If they are relevant why not other tools been used instead of them. Please justify.
Author´s Response: Thank you very much for your comments and suggestions. We used other tests such as: The Edinburgh Postnatal Depression Scale (EPDS) (The scale was validated in Spanish postpartum womenwith a sensitivity of 79% and specificity of 95.5% being its negative predictive value of 97.7%. (Garcia Esteve et al., 2003); The Birth Satisfaction Scale (BSS); This scale was used and adapted to Spanish context with internal reliability found to be acceptable. Evidence for good divergent and convergent validity was also found across total and sub-scale scores (Romero-Gonzalez et al., 2019). The main reason to use these other instruments was to test divergent validity via correlations between the newly validated MPSS with EPDS and BSS. Also, the EPDS was used in the original validation study of the MPSS to demonstrate divergent validity and is considered “the best available patient-reported screening measure of maternal postpartum depression” (Sultan et al., 2022, doi: 10.1001/jamanetworkopen.2022.14885). Postpartum stress often goes unrecognized because it is associated with specific mental conditions (such as Postnatal Depression) or even misjudged by having a negative birth experience (Fairbrother et al., 2016; Rouhi et al., 2019; Romero-Gonzalez et al., 2019).
Authors need to present in a more detailed way the relevant research in other cultural contexts both quantitative using the same tool as well as qualitative research relevant to perspectives to this topic (e.g. doi: 10.4081/hpr.2020.9178).
Author´s Response: Thank you very much for your comments and suggestions. We added the reference provided by the reviewer and expanded the context about cultural implications of maternal stress phenomenom. Lines 73-76.
Why did you choose this specific age range?
Author´s Response: Thank you very much for your comments and suggestions. We chosen this age range mainly for covering the maximum period where natality happens in Spain; the amount of women that get pregnant after the age of 45 is small considering the population that become pregnant before that age. In addition, the legal age for adulthood in Spain starts at 18 years old; and for ethical purposes we wanted to include only women above the majority of legal age to gain the informed consent. Dear reviewer you can check the mean age for maternity in Spain in the link above if you wish, we think there is no reason to include it as a citation due may be irrelevant to most of the readers. (https://www.ine.es/jaxiT3/Datos.htm?t=1579)
Is the sample representative? Please provide detailed information about the recruitment and testing process.
Author´s Response: Thank you very much for your comments and suggestions. A couple of lines were added in the methods section (Lines 93-96) in order to add more context about recruitment and testing process. We think reads better now.
The statistical analyses are naive. Please read relevant papers presenting new methods in psychometrics (https://doi.org/10.1037/met0000317)
Author´s Response: Thank you very much for your comments and suggestions. Our analysis as well as material and methods section had been supervised by a experienced statitian; Alejandro de la Torre Luque who is a lecturer in the Department of Legal Medicine, Psychiatry and Pathology at the Complutense University of Madrid. We also count in the team with Sandra Nakić Radoš and Maja Brekalo; both of them apart from their expertise in the statistical analyses, they worked in the development and creation of the tool and its first validation for clinical use. They also have checked the methods section and approved the final version of the manuscript. The paper talks about classifying individuals in classes, saying “This article discusses two statistical approaches to binary classification of respondents as diagnosed or not based on their item responses” and “These findings are not limited just to the area of diagnostic assessment, but could extend to (…) any test that assesses a construct and the process reduces to a binary decision” We think that the methods shown in this paper, although very interesting to take into consideration towards future investigations, are not applicable to the MPSS at the current point, given that, nowadays, this scale does not have any cut off points to discriminate between classes. It would be an interesting addition to consider in the future the addition of a cut point, for instance to classify in high or low risk.
The discussion must be more critical and include more relevant references.
Author´s Response: Thank you very much for your comments and suggestions. We followed international recommendations of scientific journals of health sciences (Pulido M. Internet guide on instructions to authors of more than 2000 biomedical journals: Raymon H. Mulford Library. Med Clin (Barc).1999;113:119. International Committee of Medical Journal Editors. Uniform requirements for manuscripts submitted to biomedical journals. Med Clin (Barc).1997;109: 756-63. Additional statements from the International Committee of Medical Journal Editors. CMAJ. 1997;156: 571-8.). By doing so, not only we follow the guidelines and recommendations for scientific writing, it allows to understand the impact of the findings by comparing them with other researches. This is important because help to fill the gaps that literature may not cover in the present day. We added a few references to improve the novelty of the studies cited. We hope it reads better now.
Finally, the conclusions need to be more extended and mention in a clear way why this tool is better than other existing tools for use in the Spanish population of women.
Author´s Response: Thank you very much for your comments and suggestions. We had fully rewrite the conclusion to add more emphasis about the benefits of MPSS against other tools for its use in the Spanish population of women.